# Development of frequency tuning shaped by spatial cue reliability in the barn owl's auditory midbrain

Keanu Shadron*, José Luis Peña

Dominick P Purpura Department of Neuroscience, Albert Einstein College of Medicine, Bronx, United States

**Abstract** Sensory systems preferentially strengthen responses to stimuli based on their reliability at conveying accurate information. While previous reports demonstrate that the brain reweighs cues based on dynamic changes in reliability, how the brain may learn and maintain neural responses to sensory statistics expected to be stable over time is unknown. The barn owl's midbrain features a map of auditory space where neurons compute horizontal sound location from the interaural time difference (ITD). Frequency tuning of midbrain map neurons correlates with the most reliable frequencies for the neurons' preferred ITD (Cazettes et al., 2014). Removal of the facial ruff led to a specific decrease in the reliability of high frequencies from frontal space. To directly test whether permanent changes in ITD reliability drive frequency tuning, midbrain map neurons were recorded from adult owls, with the facial ruff removed during development, and juvenile owls, before facial ruff development. In both groups, frontally tuned neurons were tuned to frequencies lower than in normal adult owls, consistent with the change in ITD reliability. In addition, juvenile owls exhibited more heterogeneous frequency tuning, suggesting normal developmental processes refine tuning to match ITD reliability. These results indicate causality of long-term statistics of spatial cues in the development of midbrain frequency tuning properties, implementing probabilistic coding for sound localization.

**\*For correspondence:** keanu.shadron@einsteinmed.edu

**Competing interest:** The authors declare that no competing interests exist.

## Editor's evaluation

This research advance shows that if juvenile barn owls experience experimentally altered interaural time differences – the binaural cue used for localizing sounds in the horizontal plane – the frequency tuning properties of neurons in the space-mapped region of the midbrain undergo adaptive changes. The results therefore suggest that the statistics of sound stimulation can influence the sensitivity of auditory midbrain neurons to a fundamental stimulus feature in the developing barn owl brain. These findings will be of interest to the fields of developmental and sensory neuroscience.

## Introduction

To accurately and efficiently perceive and react to the environmental scene, the brain must rely on the sensory cues that are naturally most reliable. The ability of the brain to quickly weigh ongoing reliability of sensory cues within different modalities has been well documented (*Jacobs and Fine, 1999*; *Rosas et al., 2005*; *Fetsch et al., 2011*; *Dacke et al., 2019*). However, in instances where sensory cue reliability is anticipated to be relatively stable and predictable, predetermined weights of sensory cues may be optimal. An example of this neural operation is particularly seen in human speech processing, where reliable phonetic properties are stable across speakers (*Holt and Lotto, 2006*; *Iverson et al., 2003*; *Toscano and McMurray, 2010*), and may only change over the course of decades (*Toscano and*

*Lansing, 2019*). While extensive research has shown changes in neural responses induced by sensory statistics (*David et al., 2004*; *Dean et al., 2005*; *Fetsch et al., 2011*), properties and fundamental mechanisms of this adaptive coding are important for understanding whether and how brain development is influenced by specific high-order statistics of sensory cues. In this report, we used the barn owl as a model organism of sound localization to test whether and how the brain develops and adapts to changes in stimulus reliability by altering the tuning properties of sensory neurons in a manner predictive of these relevant natural statistics.

The barn owl is a highly specialized species, able to hunt in the dark solely using auditory stimuli (*Payne, 1962*). Barn owls use interaural time difference (ITD), the delay for a sound to reach one ear before the other, and interaural level difference (ILD), the difference in sound intensity between the two ears, to, respectively, compute sound location in azimuth and elevation (*Moiseff and Konishi, 1981*; *Moiseff, 1989*). ITD and ILD are represented in the barn owl's external nucleus of the inferior colliculus (ICx), creating a topographic midbrain map of sound location (*Knudsen and Konishi, 1978*). The barn owl shows specialization in its ability to compute ITD from frequencies as high as 10 kHz (*Wagner et al., 1987*; *Carr and Konishi, 1990*; *Köppl, 1997*). The reliability of ITD is described as how corruptible the ITD cue is to concurrent sounds, which is determined by the acoustical properties of the head and the frequencies carrying the ITD cue (*Keller and Takahashi, 2005*; *Cazettes et al., 2014*; *Fischer and Peña, 2017*). These properties are summarized in the head-related transfer function (HRTF) (*Wightman and Kistler, 1989*; *Poon and Brugge, 1993*; *Brugge et al., 1994*; *Brugge et al., 1994*; *Hartung and Sterbing, 1997*; *Keller et al., 1998*), which describes the directional filtering that the external ears induce onto incoming sounds. These filtering properties vary across sound source location relative to the ears and the sound frequency. HRTF-based analysis has indicated that for sounds originating from frontal locations, ITD cues derived from the higher frequencies of the barn owl's hearing range are less susceptible to corruption from concurrent sounds; while for sounds from peripheral space, ITD cues derived from low frequencies are less susceptible to corruption (*Cazettes et al., 2014*; *Fischer and Peña, 2017*). Because ITD is detected by ongoing phase locking within narrow frequency bands in the lower brainstem (*Carr and Konishi, 1990*) and that the HRTF may induce instantaneous effects on stimulus shape, the reliability of interaural phase difference (IPD) is the most precise assessment of ITD cue reliability across single-frequency channels (*Fischer and Peña, 2011*; *Cazettes et al., 2014*; *Fischer and Peña, 2017*). In accordance with this, ITD reliability is considered equivalent to IPD reliability for the purposes of this report. Consistent with the hypothesis of anticipated coding of sensory cue reliability, neurons in the ICx are tuned to the frequencies that are most reliable for their preferred ITD, even if the ongoing statistics are briefly changed (i.e. using earphones which bypass the head's filtering properties) (*Cazettes et al., 2014*). A relationship between frequency tuning and ITD tuning has been reported in the midbrain and brainstem of mammalian models as well; however, this was proposed to be related to ITD neural coding and detection properties rather than anticipated ITD reliability (*McAlpine et al., 2001*; *Hancock and Delgutte, 2004*; *Day and Semple, 2011*; *Bremen and Joris, 2013*). In contrast, recent findings have shown that human spatial perception is driven by natural ITD statistics across frequencies, including ITD variability induced by concurrent sounds (*Pavão et al., 2020*), suggesting commonalities between humans and owls in the anticipation of ITD cue reliability based on acoustic properties of the head. Overall, these previous studies indicate that the predictive coding of natural ITD reliability is inherent in the brain of humans and owls, but directly testing development and causality of this statistical property on the frequency tuning of neurons representing auditory space and whether this coding is innately fixed or experience dependent remain open questions.

In the barn owl, the facial ruff, a disc of stiff feathers that surrounds the head, acts as an external ear, modulating the gain and phase of sounds reaching the eardrums (*Coles and Guppy, 1988*; *Keller et al., 1998*; *von Campenhausen and Wagner, 2006*; *Hausmann et al., 2009*). Because the filtering effects of the facial ruff are direction and frequency dependent, we sought to assess whether the removal of the facial ruff would induce a change in the pattern of ITD reliability, and whether the barn owl's auditory system adapts to these long-term changes. Previous reports studying the facial ruff removal in barn owls have focused on changes to the ILD (i.e. elevational) tuning, while noting few changes to the ITD tuning of neurons in the owl's midbrain map of auditory space (*Knudsen et al., 1984*). However, to our knowledge, there have been no studies that assessed whether ITD reliability and frequency tuning of midbrain neurons changed after facial ruff removal.

Based on these open questions, comparative analysis of ITD reliability was conducted using HRTFs from owls before and after facial ruff removal. Following this, the frequency tuning of neurons was measured in the ICx of barn owls where the facial ruff was removed during juvenile development. Additionally, the earliest-to-date recordings of the developing ICx were conducted in normal juvenile owls before the facial ruff develops but after hearing onset. We found frontally tuned ICx neurons of juvenile and ruff-removed owls were predominately tuned to frequencies lower than the observed frequency tuning in normal adult owls. These changes in both ruff-removed and juvenile owls are consistent with estimated differences in ITD reliability between normal and ruff-absent conditions. These results indicate that tuning to high frequencies of frontally tuned midbrain map neurons is developed and driven by the experience of natural auditory scenes during early life in the barn owl. In addition, recordings in the region immediately upstream of ICx confirmed that the ability to compute ITD from high frequencies (*Carr and Konishi, 1990*) was preserved in ruff-removed owls, but not inherited by ICx. Overall, this study demonstrates that the owl's sound localization pathway implements an experience-dependent representation of anticipated ITD statistics in the midbrain map of space, supporting the idea of the brain's adaptive anticipation of natural high-order sensory statistics.

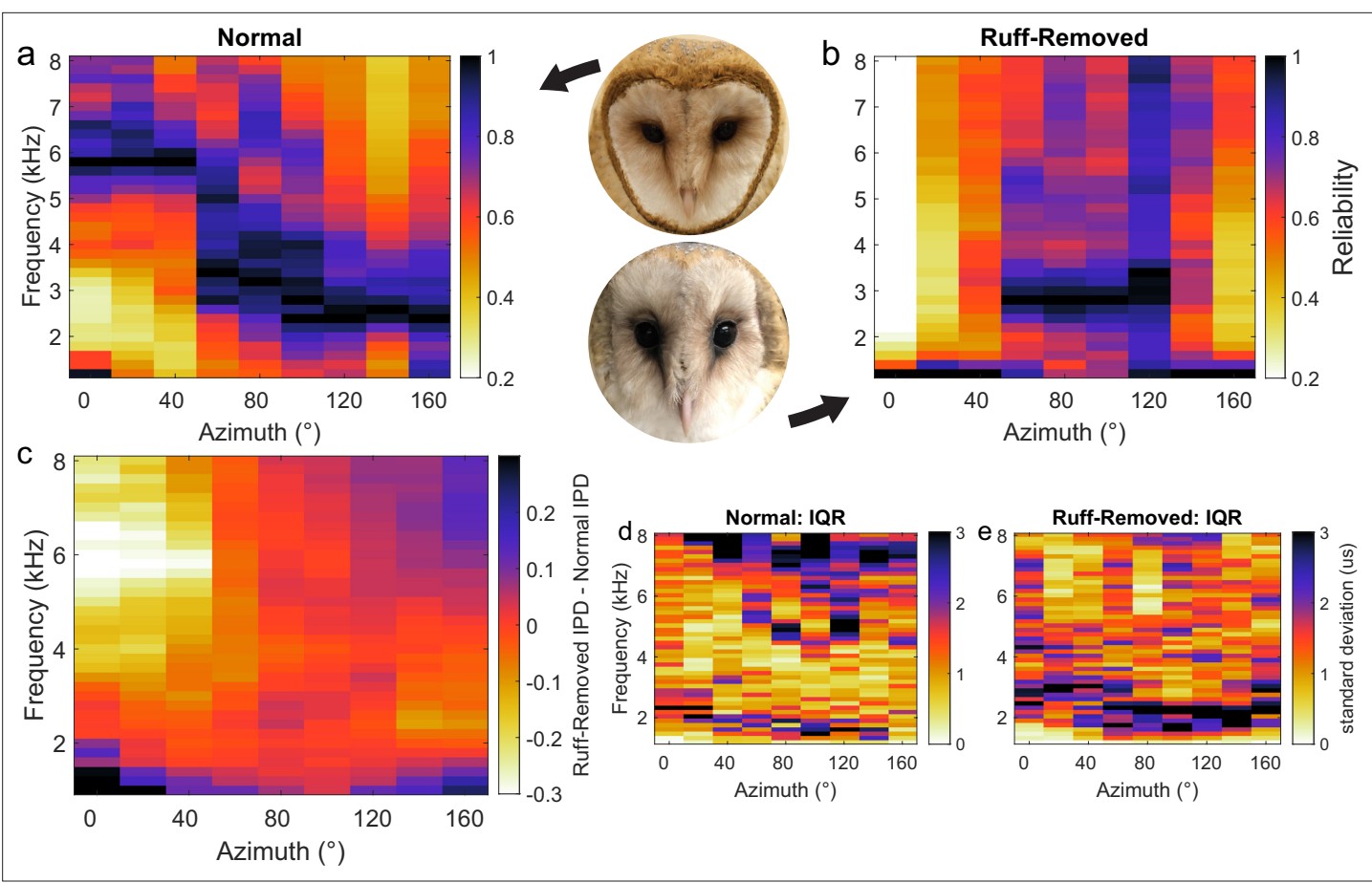

**Figure 1.** Head-related transfer function (HRTF)-based interaural phase difference (IPD) reliability in owls with and without facial ruff. Target and masker broadband sounds across varying azimuth locations were convolved with HRTFs from owls before and after facial ruff removal, then summed. IPD reliability (s.d.⁻¹) was computed across frequencies and normalized for each location, then averaged across owls (see 'Methods') before (**a**) and after (**b**) the facial ruff removal (right hemifield shown). The difference between normal and ruff-removed IPDs indicates a substantial decrease in reliability for frequencies above 4 kHz at frontal locations (**c**). HRTF data from *von Campenhausen and Wagner, 2006*.

The online version of this article includes the following figure supplement(s) for figure 1:

**Figure supplement 1.** Interaural phase difference (IPD) reliability across acoustical contexts.

## Results

To first determine how ITD reliability changes after facial ruff removal, five HRTFs from barn owls before and after facial ruff removal were used, from the dataset originally published in *von Campenhausen and Wagner, 2006*. Given ITD is computed by comparing IPD across narrow frequency channels (*Carr and Konishi, 1990*; *Fischer et al., 2011*), ITD reliability is determined by IPD reliability. Thus, for the purposes of this analysis, IPD reliability underlies ITD reliability and will be used interchangeably. A signal's phase can be corrupted by concurrent sounds, altering the IPD in a frequency-dependent manner (*Keller and Takahashi, 2005*; *Cazettes et al., 2014*; *Fischer and Peña, 2017*). For a given sound source and frequency, the amount of corruption varies with the location of the masker sound. IPD reliability can be computed as the inverse of the standard deviation of this variability (*Trommershäuser et al., 2011*; *Cazettes et al., 2014*; *Fischer and Peña, 2017*). To this end, we calculated the IPD for a given sound source across different locations of a second sound source using the HRTFs before and after facial ruff removal, on a frequency-by-frequency manner. Before ruff removal, the pattern of higher IPD reliability for high and lower frequencies in, respectively, frontal and peripheral locations (*Figure 1a*) was largely the same as previously reported (*Cazettes et al., 2014*). After ruff removal, there was a sharp decrease in the reliability of high frequencies coming from frontal locations, with minimal changes elsewhere (*Figure 1b*). Computing the difference in reliability between the two conditions highlights that after ruff removal, the changes in reliability were stronger for frequencies above 4 kHz coming from within ±40° of azimuth location (*Figure 1c*). As HRTFs can vary between individuals, we measured the variability in IPD standard deviation across owls for the

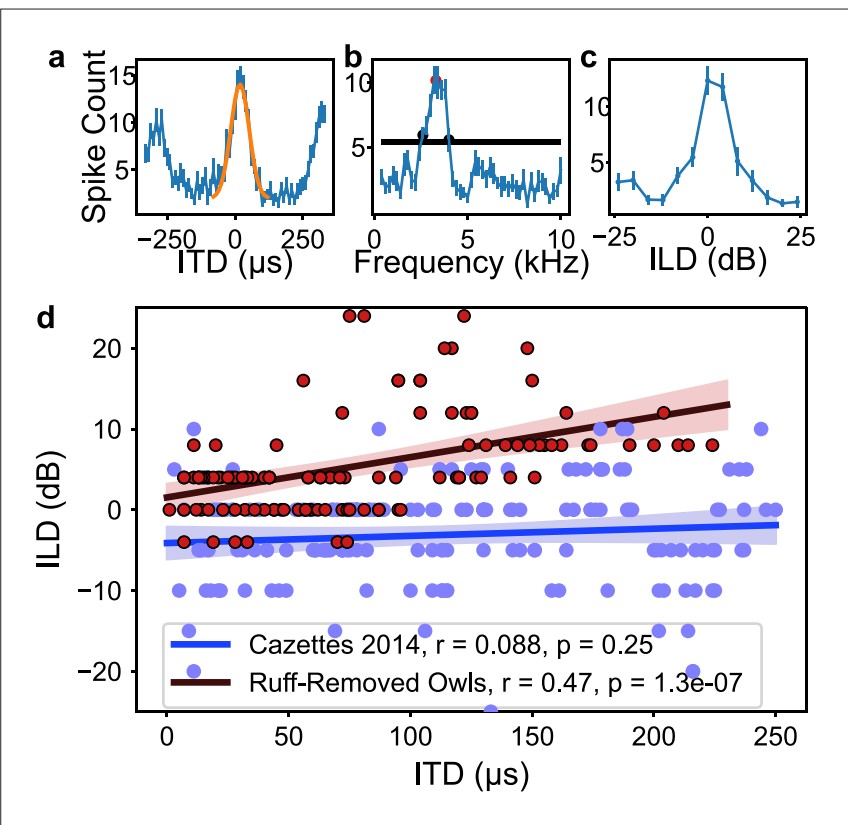

**Figure 2.** Inferior colliculus (ICx) neural responses in the ruff-removed barn owl. (**a**) Example interaural time difference (ITD) tuning curve. Yellow curve represents Gaussian fit to main peak, with the maximum of this curve termed best ITD. (**b**) Example frequency tuning curve. Black line indicates half-height used for determining half-width, low- and high-frequency bounds (black dots). Best frequency corresponds to the mean of the frequency bounds (red dot). (**c**) Example interaural level difference (ILD) tuning curve. (**d**) Best ILD plotted as a function of best ITD of neurons from ruff-removed (red dots) and normal owls (blue dots, from *Cazettes et al., 2014*). While there is no correlation between ITD and ILD tuning in normal owls (blue line), there is a correlation in ruff-removed owls (red line).

normal (*Figure 1d*) and ruff-removed (*Figure 1e*) conditions. There was notably low variability across the owls, especially at the frequencies of interest. This suggests that this measurement, based on HRTF-based analysis, leads to largely common values among individual owls. Previous reports show that there is a uniform decrease in gain of sound level across frequencies after facial ruff removal (*von Campenhausen and Wagner, 2006*), nullifying the interpretation that this is merely due to a loss of gain specifically at these higher frequencies. Repeating this analysis under different conditions, where masker sounds were quieter (*Figure 1—figure supplement 1a and b*) and using prey vocalizations as potentially natural sounds (*Figure 1—figure supplement 1c and d*), produced qualitatively similar patterns of ITD reliability, suggesting these effects are stable across potential differences of acoustic environments. These acoustical simulations suggest that if ICx frequency tuning is driven by IPD reliability, and by extension ITD reliability, then we should expect lower frequency tuning in frontally tuned ICx neurons in owls raised without the facial ruff.

To test whether frequency tuning in the barn owl's ICx is shaped by ITD reliability, we modified ITD reliability by trimming the facial ruff from two barn owls as it grew in. While previous experiments have studied the effects of physical and virtual removal of the facial ruff on the spatial tuning of ICx neurons and sound localizing behavior (*Knudsen et al., 1984*; *Hausmann et al., 2009*), none of these reported an effect on neuronal frequency tuning in ruff-removed owls. Towards this goal, once these ruff-removed owls reached adulthood (6 mo of age), we performed electrophysiological

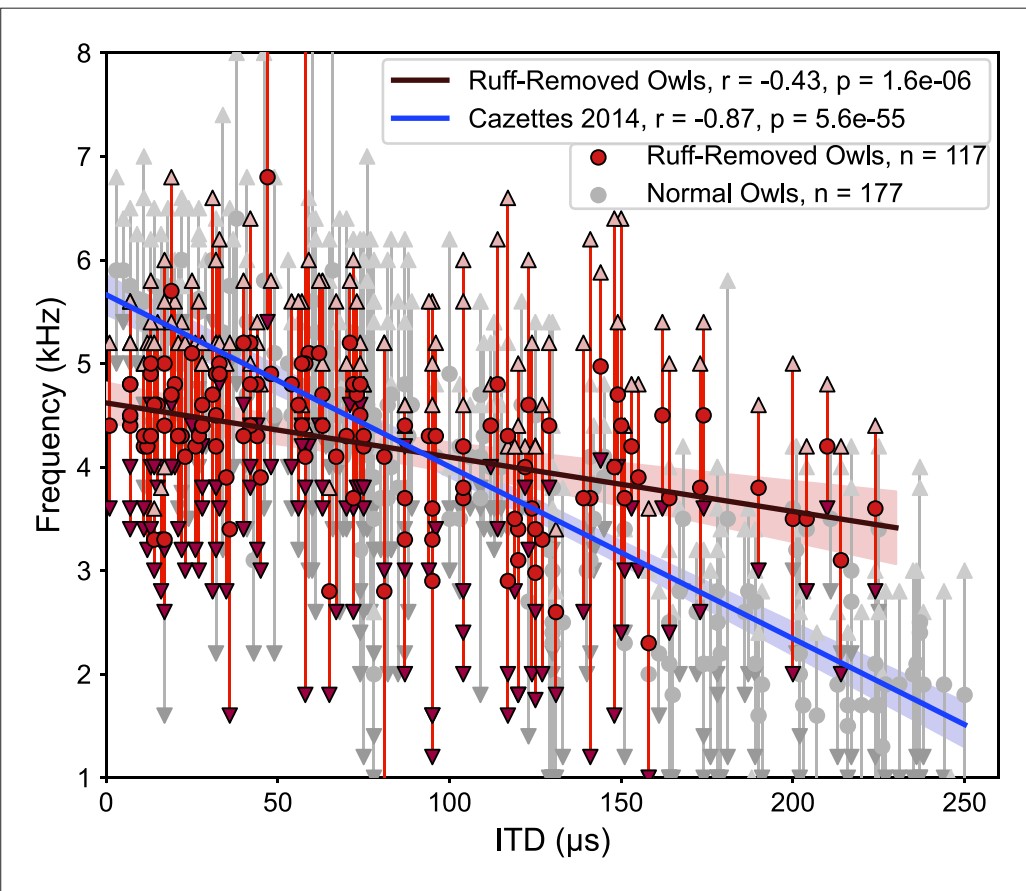

**Figure 3.** Different correlations between interaural time difference (ITD) and frequency tunings in ruff-removed and normal owls. Best frequency plotted as a function of best ITD of inferior colliculus (ICx) neurons from ruff-removed owls (red dots) and normal owls (gray dots, from *Cazettes et al., 2014*). Arrows indicate frequency bounds of each neuron; vertical lines denote frequency range.

The online version of this article includes the following figure supplement(s) for figure 3:

**Figure supplement 1.** Frequency tuning of frontally tuned inferior colliculus (ICx) neurons in ruff-removed owls.

**Figure supplement 2.** Interaural time difference (ITD) sensitivity across frequencies in the ruff-removed owl inferior colliculus (ICx).

recordings across the ICx. We recorded 117 ICx neurons across the two owls. ITD and frequency tunings were then assessed for each neuron. An example ICx unit is shown in *Figure 2a–c*. Typical ICx neurons display sharp tunings to ITD and ILD, with relatively broad frequency tuning width (>2 kHz) (*Moiseff and Konishi, 1983*; *Takahashi and Konishi, 1986*). Thus, individual ICx neurons recorded from ruff-removed owls had typical ITD and ILD tuning properties. The recorded population spanned ITD tunings of 0–230 μs, which equates to a range of 0–75° azimuth. However, on a population-wide scale, ILD tuning was correlated with ITD tuning in these owls (*Figure 2d*). This arises from the loss of bilateral asymmetry conferred by the facial ruff, which causes ILD to covary with elevation in normal conditions (*Coles and Guppy, 1988*; *Moiseff, 1989*; *Keller et al., 1998*). Consistent with previous reports, ILD covaries with azimuth after facial ruff removal (*von Campenhausen and Wagner, 2006*; *Hausmann et al., 2009*; *Knudsen et al., 1984*; *Figure 2d*).

The correlation between ITD and frequency was markedly different in the ruff-removed owls compared to normal adult owls. In normal adult owls, neurons tuned to ITDs near 0 μs, or frontal locations, are driven by high frequencies, while neurons tuned to large ITDs, or peripheral locations, are driven by low frequencies (*Cazettes et al., 2014*), exposing a significant correlation between ITD and frequency tuning ($r^2 = 0.75$, p=5.6e-55). In the ruff-removed owls, there was a shallower slope of the linear correlation between ITD and frequency tuning ($r^2 = 0.18$, p=1.6e-06, *Figure 3*). These two correlations were significantly different (*t*-test: $t = 8.97$, p=3.0e-19). The y-intercepts of the regression lines, which can be used to compare the frequency tuning of frontal neurons, were also significantly different (normal owls = 5.6 kHz, ruff-removed owls = 4.6 kHz; $t = 7.56$, p=4.0e-14). In addition, a Mann–Whitney *U*-test comparing the best frequency tuning of the frontally tuned neurons, defined here as preferring ITDs ≤ 30 μs (~10° eccentricity), produced consistent results (u = 63, p=7.3e-08). We defined best frequency as the median of a neuron's frequency range (*Figure 2b*), so this suggests that the two groups cover different frequency ranges. In line with this, the frontally tuned neurons of ruff-removed owls showed little responsivity to tones above 6 kHz (*Figure 3—figure supplement 1*). ICx neurons of ruff-removed owls showed little ITD selectivity to frequencies outside their frequency range (*Figure 3—figure supplement 2*), suggesting that these neurons cannot use high frequencies

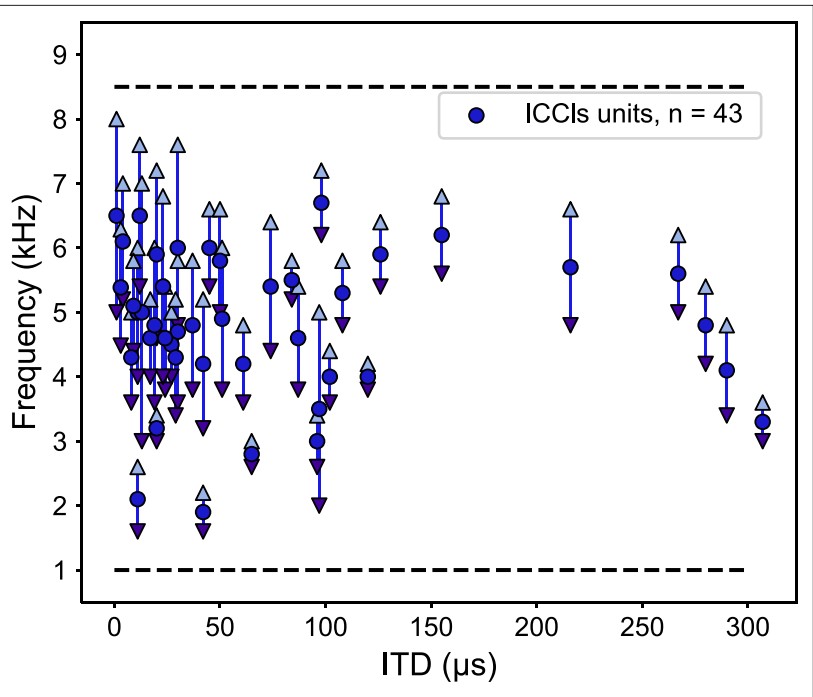

**Figure 4.** Frequency tunings in the inferior colliculus (ICCls) of ruff-removed owls span across the owl's normal hearing range. Frequency tuning of ICCls neurons of ruff-removed owls plotted as a function of their best interaural time difference (ITD). Best frequency denoted by blue dots; frequency range denoted by arrows. Black dashed lines indicate the upper and lower typical frequency range of the barn owl's ICCls (from *Wagner et al., 2007*).

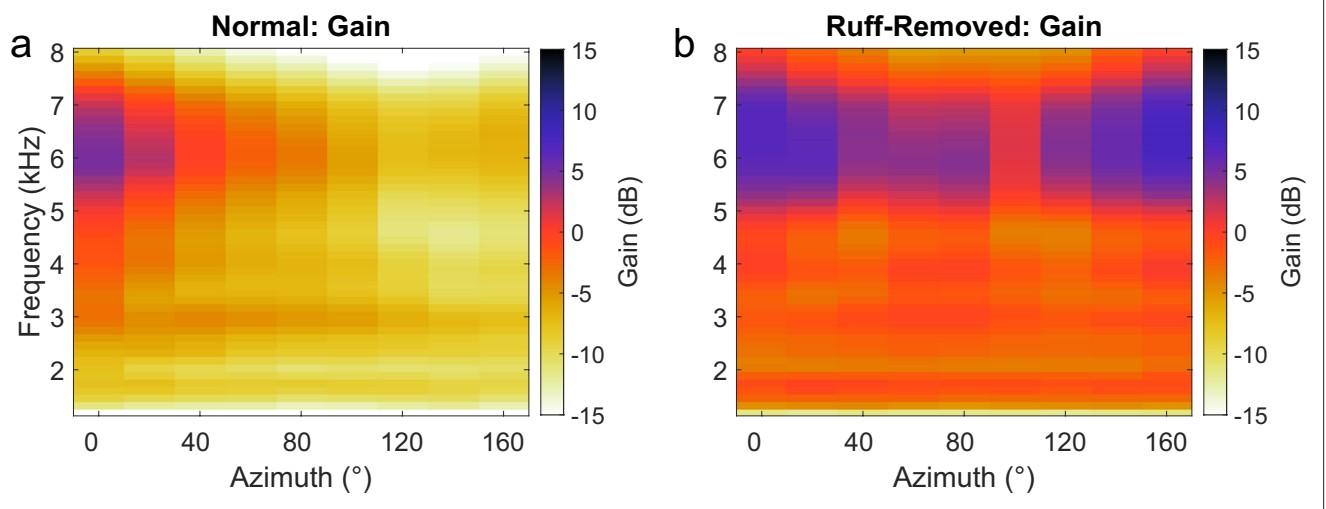

**Figure 5.** Gain across frequencies in the barn owl. Gain computed for each frequency using normal (**a**) and ruff-removed (**b**) head-related transfer functions (HRTFs). Positive gain indicates an increase in the sound level, relative to the absence of the head to filter the sounds (0 dB). Negative gain indicates a decrease in sound level.

for sound localization. The pattern of frequency tuning seen in the ruff-removed owls corresponds to the changes in ITD reliability, where the relationship between ITD and azimuth is approximately 3 µs/degrees (*Keller et al., 1998*; *von Campenhausen and Wagner, 2006*).

One potential cause of this change in frequency tuning could be due to disrupted ITD detection mechanisms induced by a change in gain of sound level, rather a change in ITD reliability. To test for this alternative, electrophysiological recordings of the lateral shell of the central nucleus of the inferior colliculus (ICCls), the immediate upstream region of ICx, were performed. ICCls neurons also display a topographic mapping of ITD, but, critically, are narrowly tuned to frequency, which increases along the dorsal-ventral axis, with tunings spanning the barn owl's hearing range of 0.5–9 kHz (*Knudsen and Konishi, 1978*; *Wagner et al., 2007*). We recorded 43 ICCls neurons from the same two ruff-removed owls in adulthood and found the normal frequency tuning range (1–8 kHz) across ITD-selective neurons (*Figure 4*). In particular, ICCls neurons tuned to high frequencies and frontal ITDs in ruff-removed owls were observed (*Figure 4*). This suggests that the auditory system is still able to use high frequencies to compute ITDs, but this is not passed onto the ICx.

In addition, changes in directional gain were considered as an alternative cause driving frequency tuning changes in the ruff-removed condition. Towards this end, the gain across frequency and location was computed from normal and ruff-removed HRTFs. In line with previous reports (*Keller et al., 1998*; *Cazettes et al., 2014*), gain in the normal condition was strongest for high frequencies in frontal space, with negative gain at all other locations and frequencies (*Figure 5a*). In the ruff-removed condition, gain remained strongest for high frequencies in frontal space, but expanded slightly to include stronger gain in more peripheral locations (*Figure 5b*). In contrast, the gain of lower frequencies had a more limited increase, to approximately 0 dB (i.e. no net gain change), likely because the facial ruff normally attenuates these sounds (*Figure 5a*, *Coles and Guppy, 1988*; *Keller et al., 1998*). These findings are consistent with previous analysis of changes in gain across frequencies following ruff removal (*von Campenhausen and Wagner, 2006*). This analysis suggests that the limited changes in gain across frequencies are insufficient to explain the changes in frequency tuning as high frequencies remain the most audible in frontal locations after facial ruff removal.

Alternatively, the change in the frequency tuning of frontal neurons could be due to a remapping of ITD and ILD in a frequency-dependent manner following facial ruff removal. To test this possibility, we modeled the spatial tuning of frontally tuned neurons across frequencies. Based on HRTFs from *von Campenhausen and Wagner, 2006*, the ITD and ILD was computed for each spatial location sampled, across frequencies (described in 'Methods' section). These ITDs and ILDs were passed through a model ICx neuron tuned to 0 µs ITD and 0 dB ILD to simulate the neuron's response across space (*Figure 6a*). This analysis was performed using the HRTFs of owls before (*Figure 6b*) and after

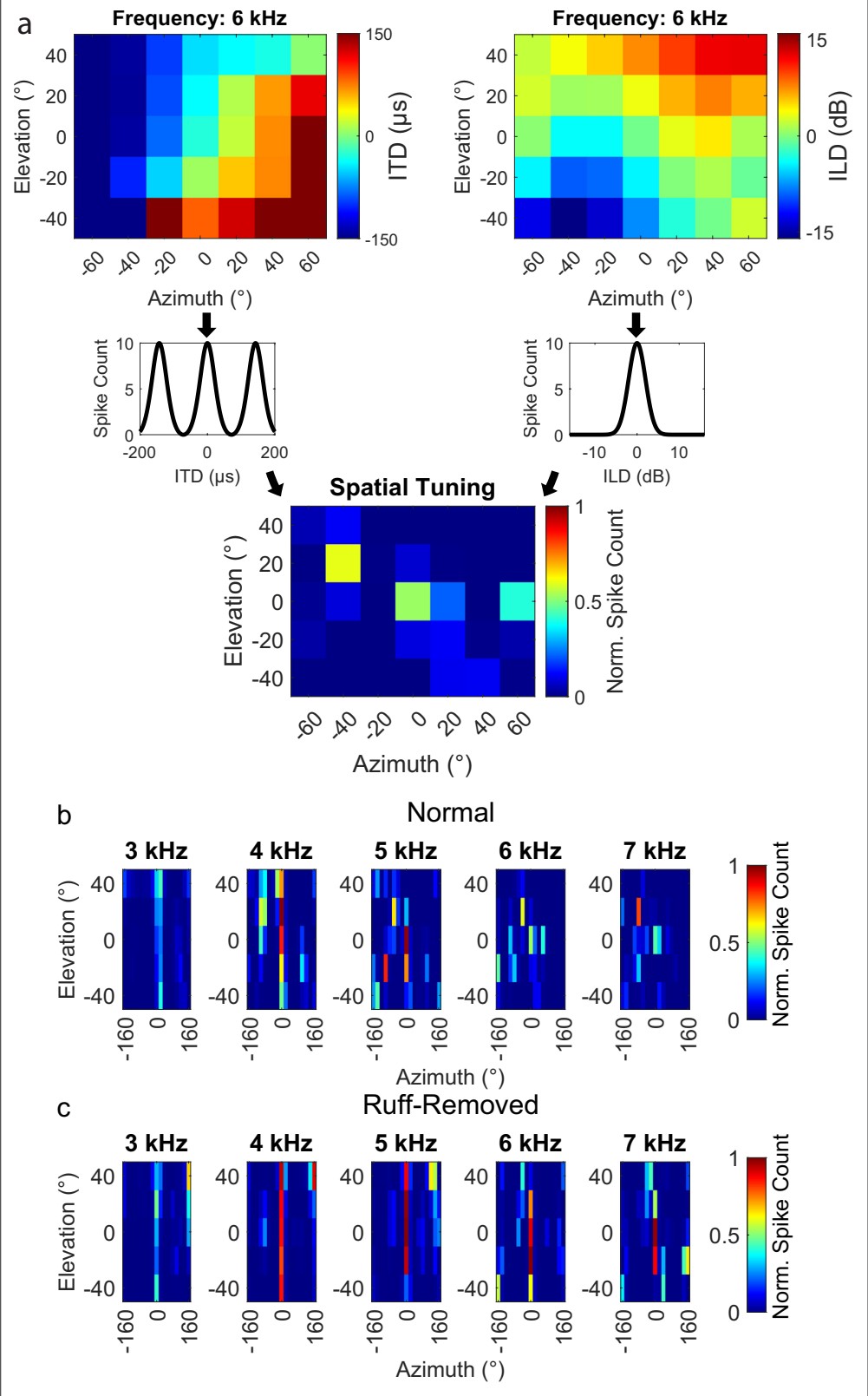

**Figure 6.** Spatial tuning for frontal space following ruff removal does not show systematic alterations based on frequency. (**a**) Schematic methodology to determine spatial tuning, using 6 kHz as an exemplary tone. Interaural time difference (ITD) (upper left) and interaural level difference (ILD) (upper right) were estimated as a function of the spatial location of the sound source. Neural responses were predicted by entering the estimated ITD and

*Figure 6 continued on next page*

*Figure 6 continued*

ILD into modeled tuning curves for a simulated frontally tuned neuron (middle plots). The overall spatial tuning was calculated by combinatorial multiplication then normalization of these modeled responses (lower plot). (**b, c**) Spatial tuning maps for owls before (**b**) and after (**c**) ruff removal, displayed for five example tones.

(*Figure 6c*) facial ruff removal. In the ruff-removed condition, spatial tuning to elevation widened dramatically, as expected from previous reports (*Knudsen et al., 1984*; *von Campenhausen and Wagner, 2006*; *Hausmann et al., 2009*). However, we did not find any widening of modeled azimuthal spatial tuning nor any acoustical loss in the potential efficacy of high frequencies to compute ITD. In fact, after facial ruff removal, we noticed an increase in the potential efficacy of high frequencies for frontal space, likely due to ILD being correlated with azimuth after facial ruff removal (*Knudsen et al., 1984*; *Figure 2d*). These results further support the hypothesis that the barn owl's frequency tuning is driven by ITD reliability rather than solely spatial tuning.

We next sought to determine the frequency tuning of juvenile owls before the facial ruff fully developed. Recordings were performed from two juvenile owls at 42 and 44 d post hatching. These time points corresponded to ongoing ruff development as the ruff feathers were still within the sheathed stalks, with the ruff not fully developing until approximately 60 d post hatching (*Haresign and Moiseff, 1988*). To our knowledge, these recordings are the earliest performed during the development of the barn owl's midbrain. We were able to identify 39 ICx neurons, which showed typical topographic ITD tuning, as well as latencies and thresholds. However, we found that ILD tuning did not change with depth along the dorsal-ventral axis, as observed in normal adult ICx (*Figure 7—figure supplement 1*; *Mogdans and Knudsen, 1993*). Because the relationship between ILD and elevation is determined by the facial ruff (*Knudsen et al., 1984*; *von Campenhausen and Wagner, 2006*), this provided further evidence that the facial ruff was still underdeveloped in these juvenile owls. There was no

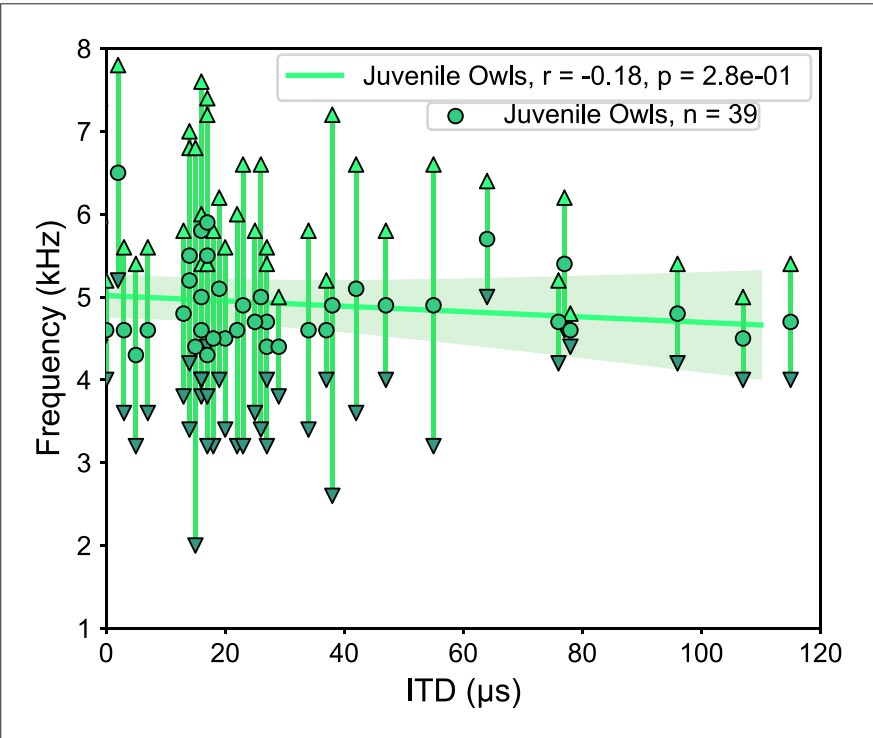

**Figure 7.** Tuning properties of the juvenile owl's inferior colliculus (ICx) neurons. ICx neurons recorded from juvenile owls, before the facial ruff developed, plotted by their best interaural time difference (ITD) and frequency tuning range. Weak correlation between ITD and frequency (green line).

The online version of this article includes the following figure supplement(s) for figure 7:

**Figure supplement 1.** Correlated interaural level difference (ILD) and interaural time difference (ITD) tuning in the juvenile owl inferior colliculus (ICx).

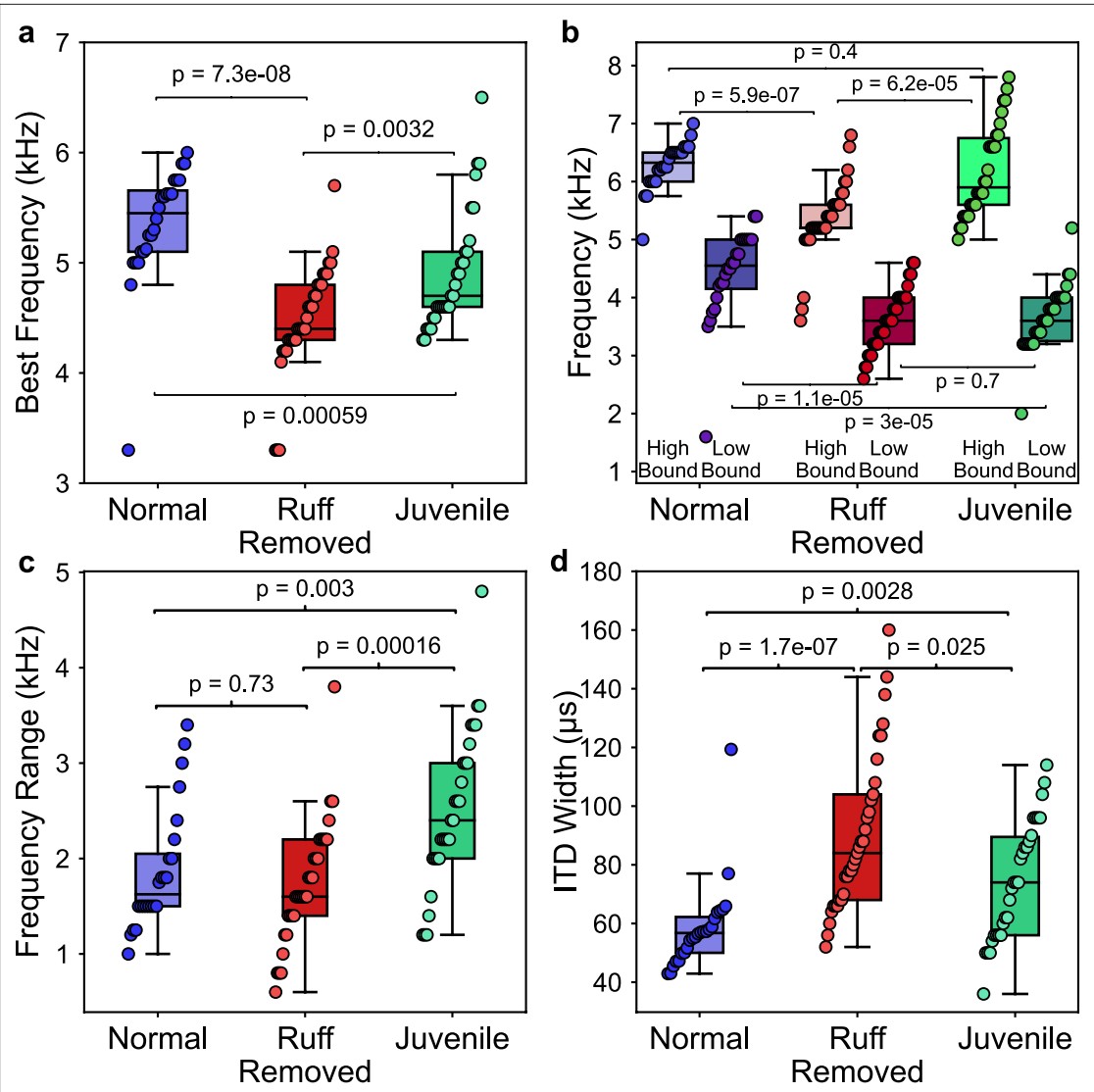

**Figure 8.** Comparisons of frequency and spatial tunings of frontally tuned inferior colliculus (ICx) neurons between normal, ruff-removed and juvenile owls. (**a**) Box plots indicating the distribution of best frequencies for neurons tuned to frontal interaural time differences (ITDs) (±30 μs, equivalent to approximately ±10° azimuth) for normal (blue, n = 24), ruff-removed (red, n = 33) and juvenile (green, n = 30) owls . (**b**) Box plots for each group, denoting the high and low bounds of frequency tuning curves. (**c**) Box plots representing the responsive frequency range for neurons of each group. (**d**) Box plots representing the width of ITD tuning curves of each group. Individual neurons and Mann–Whitney *U*-test p-values used to compare groups are shown over each box plot.

correlation between ITD and best frequency in the juvenile owls ($r^2$ = 0.18, p=0.28), although this may be attributed to the lack of recorded neurons tuned to ITDs >120 μs (*Figure 7*). Because this correlation was not significant, we confined our subsequent analyses to only frontally tuned neurons (best ITDs ≤ 30 μs) as the clearest differences between normal and ruff-removed conditions occurred in frontal space for both ITD reliability and frequency tunings. When we compared the best frequency tuning of frontal neurons, there was a significant difference between the juvenile owls and both the normal (u = 558, p=5.9e-04) and ruff-removed (u = 281, p=3.2e-03) adult owls (*Figure 8a*). The mean best frequency of the frontal neurons in juvenile owls was 4.9 kHz, which is between that of normal and ruff-removed adult owls. The size of these differences between juvenile owls and both normal and ruff-removed owls, measured by biserial rank correlation, further supported this result (*r* = 0.55 and *r* = 0.43, respectively). With a mean best frequency of 4.9 kHz for frontally tuned neurons, this suggests that the neurons in the ICx of juvenile owls do not begin development tuned primarily to high frequencies, but that this develops as the facial ruff develops. Later recordings done in these same birds' ICx,

performed in the 160–200-day-old age range, display the typical high-frequency tuning for frontally tuned neurons (data not shown), supporting this hypothesis.

While there were clear differences in best frequency across juvenile, ruff-removed and normal adult owl groups, we found additional differences when comparing the high and low boundaries of each neuron's frequency tuning (see *Figure 2b* for definitions). While there was a significant difference in the neurons' high-frequency bounds between the juvenile and ruff-removed owls (u = 206, p=6.2e-05), there was no significant difference between the juvenile and normal adult owls (u = 409, p=0.4). In contrast, we saw the opposite for the low-frequency bounds: there was a difference between the juvenile and normal adult owls (u = 600, p=3e-05), but no significant difference between the juvenile and ruff-removed owls (u = 466, p=0.7). These results are summarized in *Figure 8b* and suggest that the juvenile owls' ICx neurons are broader and more heterogeneously tuned to frequency than normal or ruff-removed adult owls, which then gets refined during development. In accordance with this, we found significant differences in the responsive frequency range of ICx neurons between the juvenile owls and both the normal adult owls (u = 190, p=0.003) and ruff-removed adults (u = 221, p=1.6e-04), but not between the normal and ruff-removed adults (u = 418, p=0.73) (*Figure 8c*). In addition, because there was no difference in the responsive frequency range between the adult groups, we can deduce that their neurons are sensitive to different frequencies based on the results from *Figure 8a*. Analysis of ITD tuning width of frontal neurons showed significantly broader tuning width in ruff-removed (u = 72, p=1.2e-07) and juvenile owls (u = 188, p=0.0028), consistent with their tuning to lower frequencies and the known relationship between frequency tuning and ITD tuning width of midbrain neurons (*Takahashi and Konishi, 1986*; *Wagner et al., 2007*; *Figure 8d*). These results indicate that the juvenile owls show larger heterogeneity and broadening of frequency tuning in the ICx, which adapts based on the pattern of ITD reliability that the bird experiences along early life.

## Discussion

We report that in barn owls that developed without a facial ruff, there is an altered relationship between the tunings to ITD and frequency relative to previous reports from normal adult barn owls (*Knudsen et al., 1984*; *Cazettes et al., 2014*). These ruff-removed barn owls showed on average a decrease in frequency tuning for frontally tuned ICx neurons, from 6 kHz to 4.5 kHz, with no change in frequency tuning width (*Figure 8c*). This is consistent with the analysis of the effect of acoustical properties of the owl's head on ITD statistics. These observed changes are despite the overrepresentation of high frequencies in the cochlea (*Köppl, 1997*) and adaptations to support phase locking at these high frequencies (*Sullivan and Konishi, 1984*; *Carr and Konishi, 1990*; *Köppl, 1997*), which are extremely unlikely to change following ruff removal. In support of this, neurons immediately upstream of the ICx maintained the ability to resolve ITD at these high frequencies (*Figure 4*). The changes in ITD reliability and correlated frequency tuning support the hypothesis of an adaptive predictive coding of this ITD statistic in the owl midbrain along normal developmental and experimentally altered changes in the acoustical properties of the head.

The earliest recordings of the barn owl's midbrain, conducted in this study at 42 and 44 d after hatching, showed neuronal tuning to ITD and ILD (*Figure 7*). This is in line with previous reports on the auditory brainstem which indicate that the circuitry underlying ITD and ILD tuning are present (albeit immature) and that these binaural spatial cues are audible to the birds across their hearing range by approximately 20 d post hatching (*Carr and Boudreau, 1996*; *Köppl and Nickel, 2007*; *Kraemer et al., 2017*). However, consistent with reports of disrupted ILD tuning after facial ruff removal (*Knudsen et al., 1984*), ILD tuning was correlated with azimuthal tuning in the juvenile owls (*Figure 7—figure supplement 1*). This ILD tuning pattern indicates that the facial ruff was underdeveloped at this time, thus not contributing to auditory processing. The best frequency tuning we found for frontally tuned ICx neurons in juvenile owls were lower than what has been found in normal adults (*Cazettes et al., 2014*), but higher than what is found in ruff-removed owls (*Figure 8*). This suggests that at this time point the barn owl's ICx frequency tuning is not yet shaped by ITD reliability, which may change rapidly in response to the facial ruff's development. Consistent with this hypothesis, we found that juvenile ICx neurons had slightly wider and heterogenous frequency tuning widths than either the normal adult or ruff-removed adults, covering the ranges of both adult groups, but not a width that would indicate that each of these ICx neurons were sampling the entire frequency range available to the barn owl for ITD detection (0.5–9 kHz). This suggests high heterogeneity in the

juvenile owl's ICx frequency tuning at the time that the facial ruff would begin to grow in. As the owl develops, these individual neurons may sharpen their frequency tuning to match the pattern of ITD reliability the owl experiences during development or the neurons that respond to unreliable frequencies could be pruned during development. Future work dissecting developmental and experience-dependent changes to the ICCls-ICx synapses may elucidate the fundamental mechanisms underlying this adaptive coding.

These results suggest a juvenile owl undergoes a process to refine the frequency tuning in the ICx based on the pattern of ITD reliability that the owl experiences during early-life development. Along normal development, the frequency tuning of frontally tuned neurons changes in a manner consistent with an increase in high-frequency reliability at frontal locations, determined by the acoustic properties of the facial ruff. It should be noted that the difference in frequency tuning of frontally tuned neurons between normal adults and juveniles (rank-biserial correlation, $r = 0.55$) is smaller than the difference between normal adult and ruff-removed owls (rank-biserial correlation, $r = 0.84$). This suggests that the ruff-removed owls still underwent a refinement in ICx frequency tuning, but toward lower frequencies compared to normal adults. The timing of ruff development, which completes at approximately 60 d post hatching, aligns with previous reports of a critical period in the barn owl, which can remain open to 200 d post hatching (*Knudsen et al., 1984*; *Brainard and Knudsen, 1993*; *Brainard and Knudsen, 1998*). Although not directly tested in this study, it is likely that the shift in frequency tuning occurs over this time period.

It is unlikely that the effect of ruff removal on ITD reliability is due solely to a change in gain of sound level as this and previous reports found that ruff removal decreases the gain across frequencies, yet the gain remains higher for higher frequencies around 6–8 kHz (*Figure 5*, *Coles and Guppy, 1988*; *von Campenhausen and Wagner, 2006*), inconsistent with the estimated lower ITD reliability of high-frequency sounds in frontal locations for these owls. Likewise, the rate of change of ITD (µs/deg), another significant statistic determining ITD discriminability (*Feddersen et al., 1957*; *Gelfand, 2016*; *Pavão et al., 2020*), is not reported to change significantly across frequencies after ruff removal (*von Campenhausen and Wagner, 2006*), reducing the possibility of it being an alternative mechanism driving change of frequency tuning. This is consistent with previous work which reported that gain alone cannot explain the frequency tuning in the ICx (*Cazettes et al., 2014*). Additional testing of whether the observed change in frequency tuning of frontal neurons could be determined by the reported changes in the pattern of ITD and ILD across spatial locations after ruff removal (*Knudsen et al., 1984*; *von Campenhausen and Wagner, 2006*) showed no systematic frequency-dependent effects (*Figure 6*). While ruff removal enables the use of ILD to determine azimuthal location, we do not see a reason that this would induce a preference to lower frequencies in these neurons. Because of the shadowing effect of the head, high frequencies induce larger ILDs, increasing its resolution. Following ruff removal, the rate of change of ILD decreases across frequencies, decreasing its resolution of spatial location (*von Campenhausen and Wagner, 2006*). Thus, high frequencies would still be preferred for ILD tuning. In addition, ITD has sharper resolution than ILD overall, but especially at higher frequencies (*Coles and Guppy, 1988*; *Keller et al., 1998*; *von Campenhausen and Wagner, 2006*). Thus, spatial resolution does not explain the changes seen in frontally tuned ICx neurons. These conclusions support the premise that ITD reliability is a primary determinant of ICx frequency tuning.

Nevertheless, there may be limits to the extent that frequency tuning can be shifted due to ITD reliability. The majority of the barn owl's basilar papilla responds to 5–10 kHz (*Köppl et al., 1993*), which may induce a bias toward high frequencies into the downstream auditory centers. The eardrum itself may also have detection biases for frequencies above 3 kHz (*Kettler et al., 2016*). Consistently, the ICCls and midbrain maps contain few neurons tuned to frequencies below 1 kHz (*Figure 4*, *Wagner et al., 2007*; *Cazettes et al., 2014*). While these properties are not expected to be affected by the facial ruff, they likely evolved simultaneously with the evolution of the facial ruff to facilitate the use of ITD detection at high frequencies. This is akin to reports that the barn owl's sound localization cannot fully adapt to large displacements in the visual field (*Knudsen and Knudsen, 1989*; *Knudsen and Knudsen, 1990*).

While there were some changes in ITD reliability and gain for the most rearward located sounds (±160°) determined by HRTF-based analysis, we do not believe this would influence reported results focused on frontal and peripheral space. The ICx, as well as the downstream optic tectum, mostly cover 30° ipsilateral to 90° contralateral space, which equates to approximately +90 µs to –300 µs ITD

(**Knudsen, 1982**; **Knudsen, 1984**; **Cazettes et al., 2014**). This means that the most rearward locations are significantly underrepresented in the auditory midbrain. This is expected as the auditory space map in the ICx is projected onto the visual space map in the optic tectum, so there is no obvious visual equivalent for rearward space. Because the barn owl is binocular and relies on visual and auditory cues to locate and capture prey, the representation of rearward space in the auditory midbrain, which drives the motor response, would be unnecessary.

There has been extensive research on how the brain encodes and adapts to changes of sensory statistics. Reports of early-life exposure to low signal-to-noise ratios have suggested a delay in the maturation of sensory systems (**Withington-Wray et al., 1990**; **Chang and Merzenich, 2003**; **Efrati and Gutfreund, 2011**). In the auditory system, numerous reports showed that tonotopic maps can be adjusted in response to frequency-specific noise (**Zhang et al., 2001**; **Noreña et al., 2006**; **Villers et al., 2007**) or notches (**Gold and Knudsen, 2000**). More recent findings suggest that certain noise contexts can alter tuning based on the stimulus statistics (**Cruces-Solís et al., 2018**; **Homma et al., 2020**). While these previous reports demonstrated changes in the auditory system induced by altering the statistics between signal and noise that mediate spectrotemporal discrimination, this study further finds evidence of adaptive plasticity to statistics of the ITD sensory cue used for sound localization. Specifically, this study demonstrates that frequency tuning in the midbrain space map may also be driven by how well these frequencies convey a relevant spatial cue, ITD, which is initially computed in the brainstem (**Carr and Konishi, 1990**). This suggests that high-order statistics are detected across frequencies and locations and that adaptive coding is implemented in the barn owl's midbrain, where a map of auditory space emerges, suggesting that natural statistics are driving the implementation and development of a computational brain map (**Knudsen et al., 1987**). While the ongoing reliability of ITD cue is known to affect sound localization (**Jeffress et al., 1962**; **Saberi et al., 1998**), these findings support the theory of an essential role of anticipated ITD reliability, where its effect on neuronal tuning is acquired during development and maintained in spite of ongoing statistics, in the coding of sound location by population responses in the owl midbrain map (**Fischer and Peña, 2011**; **Cazettes et al., 2014**; **Fischer and Peña, 2017**; **Ferger et al., 2021**). In addition, the broader ITD tuning of frontal neurons in juvenile and ruff-removed owls (**Figure 8d**) was predicted by their tuning to lower frequencies and potentially consistent with a proposed theory that the shape of tuning curves encodes reliability in the midbrain map (**Cazettes et al., 2016**). The changes in frequency tuning we found occurred in animals maintained in regular group aviaries with daily feeding and absence of specific task training or survival needs such as hunting experience or auditory-driven tasks. Thus, the shaping of the owl's ICx frequency tuning based on anticipated ITD reliability may be a normal and robust step in owl's development. Because of this, the ruff-removed owl may be a useful model for future investigation of the molecular and synaptic mechanisms underlying experience-dependent coding of anticipated cue reliability.

In sum, this work provides evidence that the brain can assess the reliability of an auditory spatial cue along normal development and experience-dependent changes and use this information to drive neural tunings matching anticipated reliability across frequency and locations. To our knowledge, this is the first description of the natural learning of cue reliability, where the learning is implemented in a midbrain network, in contrast with previous work finding similar results in the forebrain (**Keating et al., 2013**). As anticipated ITD cue reliability has also been shown to drive spatial discriminability in humans (**Pavão et al., 2020**), it is possible that a similar learning process could occur as the human head and ears grow and develop. In addition, previous reports suggest the capability of the human auditory system to compute (**Saberi and Petrosyan, 2005**) and infer high-order statistics (**McWalter and McDermott, 2019**), suggesting commonalities across species. Overall, this study supports causality of anticipated sensory cue statistics of spatial cues on neuronal tuning properties in the auditory system and developmental and experience dependent plasticity driving this statistical inference.

## Methods
### HRTF analysis
Pairs of HRTFs from five barn owls, with and without the facial ruff, were provided by Dr. Wagner (**von Campenhausen and Wagner, 2006**). The computation of estimated ITD reliability based on these HRTFs was conducted using a previously reported method (**Cazettes et al., 2014**). Briefly, two

broadband signals, a target and masker, with flat spectrum across the owl's ITD detection frequency range (0.5–10 kHz; *Köppl, 1997*; *Carr and Konishi, 1990*), were convolved with head-related impulse responses at each sampled sound source location of available HRTF datasets (azimuth: ±160° at 20° steps, elevation: 0°). The summed target and masker sounds were passed through a gammatone filter bank (1–8 kHz, 0.2 kHz steps) as previously described (*Fischer et al., 2009*). Due to the sharp frequency selectivity of ITD detection neurons (*Carr and Konishi, 1990*; *Fischer et al., 2011*), IPD of narrow frequency bands, which ITD is computed from, was used for this analysis to more accurately analyze the effect of underlying acoustics across conditions. Additionally, because ITD is derived from IPD by carrier frequencies, this precludes the ability to meaningfully compare ITD reliability across frequency. As such, IPD reliability was used instead, with the term equating to ITD reliability for the purposes of this report. IPD was calculated for each frequency as the point of the maximum cross-correlation between the left and right inputs. To estimate reliability of ITD cues across frequency and locations, the circular standard deviation of IPD was computed for each target location, across all frequency ranges and masker locations. This procedure was repeated 10 times for each individual HRTF, and the circular standard deviation of IPD was averaged across repetitions. IPD reliability was calculated as the inverse standard deviation of the IPD (*Trommershäuser et al., 2011*). This procedure was repeated for alternate stimulus conditions. In the first condition, maskers were held at 0.5 the amplitude of the target sound. In the second condition, a 100 ms excerpt of a mouse scream, which has spectral power across the owl's hearing range, was used as both the target and masker sounds (recording from *Netser et al., 2011*).

For testing whether observed changes in neuronal tuning properties could be due to changes in spatial tuning, rather than ITD reliability, the same HRTFs from owls before and after ruff removal were used to compute the ITD and ILD across frequency and spatial locations sampled in these HRTFs. This analysis was conducted for tones within the range of frequencies where changes in tuning were observed (3–7 kHz, at 1 kHz steps) to estimate changes in ITD/ILD mappings for each tone. Model neuronal responses tuned to frontal space were constructed using the following equations for ITD and ILD, respectively:

$$a_n\left(ITD\right) = a_{max}\frac{\left(e^{\cos\left(2\pi f\left(ITD - \mu_n\right)\right)} - e^{-1}\right)}{e^1 - e^{-1}}$$

$$a_n\left(ILD\right) = a_{max}e^{-\frac{\left(ILD - \delta_n\right)^2}{\sigma}}$$

where f is the tone frequency (in Hz), $\sigma$ is the width parameter for the ILD tuning. The neuron's best ITD, $\mu_n$, and best ILD, $\delta_n$, were defined as 0 µs and 0 dB, respectively. The maximum spike rate for both tuning curves was 10 spikes per second, set by $a_{max}$. The spatial responses for ITD and ILD were then multiplied, as consistent with previously reported mechanisms of ICx neurons (*Peña and Konishi, 2001*) and normalized. This resulted in a full spatial tuning map of a frontally-tuned neuron across frequencies.

## Facial ruff removal

Two juvenile barn owls were hand-raised (one starting at 14 days old and one at hatching). Facial ruff was trimmed as it developed, starting at approximately 1 month old. As they were comfortable with humans, trimming occurred as owls rested, without the need of anesthesia or stress. Surgeries were not performed on these owls until 3 mo of age, approximately 1 mo after sexual maturity.

## Surgery

Data were collected in two adult owls that had undergone facial ruff removal (see above) and two normal juvenile owls. Surgeries of adult owls were performed as previously described (*Wang et al., 2012*; *Ferger et al., 2021*). Briefly, owls were first anesthetized with intramuscular injections of ketamine hydrochloride (Ketaset; 20 mg/kg) and xylazine (AnaSed, 4 mg/kg), then injected with lactated Ringer's solution (10 mL, s.c.). Supplemental injections of ketamine and xylazine were provided every 1–2 hr as needed to maintain sedation. In a preliminary surgery, a stainless-steel head-plate and reference post were attached to the skull using dental acrylic. Craniotomies were performed within dental acrylic wells above the coordinates for the auditory midbrain. During recordings, a small incision is made in the dura mater to insert electrodes. At the end of each recording, a sterile piece of

plastic was fitted to cover the craniotomy, which was then covered in a fast-curing silicone elastomer (polyvinyl siloxane; Warner Tech Care). The owl was then administered carprofen (Rimadyl; 3 mg/kg, i.m.) to reduce pain. Recordings were repeated every 14 d or longer to allow proper recovery.

Because of the development of the skull precluding early surgeries in juvenile owls, the head-plate implantation surgery and electrophysiology recording were performed on the same day in these animals. Recording surgeries were performed at 42 and 44 d post hatching. At this time point, the juvenile barn owls are slightly above normal adult weight (*Haresign and Moiseff, 1988*), requiring administered anesthetics dosages similar to adults. Both owls recovered and were standing within 1 hr of surgery completion, and fully developed normally. Subsequent recordings were performed between 130 and 190 days old to verify normal frequency tuning in adulthood (observed but data not shown).

All procedures complied with the National Institute of Health guidelines and were approved by the institutional animal care and use committee of the Albert Einstein College of Medicine.

## Acoustical stimuli

All experiments were performed in a sound-attenuating chamber (Industrial Acoustics). Stimuli were generated by System II hardware (Tucker-Davis Technologies), controlled by a computer with custom-created software. Dichotic stimuli were presented through custom-built earphones containing a speaker (model 1914, Knowles) and microphone (model EK-23 024, Knowles). Earphones were calibrated to adjust for irregularities in phase and amplitude across the owl's hearing range, from 0.5 to 13 kHz.

Acoustic stimuli consisted of either broadband (0.5–11 kHz) or tonal signals with a 100 ms duration and 5 ms rise-fall time. For tonal stimuli, a step size of 200 Hz was used, between 0.5 and 10 kHz. A 300 ms inter-stimulus interval was used to prevent response adaptation effects (*Singheiser et al., 2012*; *Ferger et al., 2018*). All stimuli were binaural and presented at least 10 dB above the neuron's threshold. To present ILD, each earphone was increased or decreased equally, such that the sound level was maintained across ILDs.

## Electrophysiological recordings

Single units were recorded extracellularly using 1 MΩ tungsten electrodes (A-M Systems) and amplified by a DP-301 differential amplifier (Warner Instruments). Electrical traces were recorded by System II hardware (Tucker-Davis Technologies) and custom-made software. The ICx and ICCls midbrain regions were targeted stereotaxically (*Knudsen and Knudsen, 1983*). In addition, neurons were distinguished based on response properties. Single units were isolated based on the spike amplitude and the uniformity of spike shape. Nevertheless, neurons close to each other in the map are known to have similar tuning properties (*Cazettes et al., 2014*). Specifically, ICx neurons display tuning to both ITD and ILD, with side peaks in ITD tuning occurring at the reciprocal period of the unit's best frequency (*Takahashi and Konishi, 1986*; *Fujita and Konishi, 1991*; *Wagner et al., 2007*; *Cazettes et al., 2016*). The response rate of these side peaks was usually suppressed as compared to the main peak (*Takahashi and Konishi, 1986*; *Mazer, 1998*; *Peña and Konishi, 2000*). Recordings in ICCls were determined by their known change in the preferred frequency tuning of neurons: along the dorsal-ventral axis, tuning to frequency increases with depth (*Knudsen and Konishi, 1978*; *Takahashi et al., 1989*; *Wagner et al., 2007*). Additionally, ICCls neurons display broad contralateral ILD tuning and side peaks in ITD tuning that are indistinguishable in response rate to the main peak (*Takahashi et al., 1989*; *Fujita and Konishi, 1991*; *Adolphs, 1993*; *Wagner et al., 2007*). A minimum of 200 μm between units was used to ensure isolation of subsequently recorded neurons.

## Data analysis

For each stimulus parameter, a response curve was computed by averaging the firing rate across stimulus repetitions. A Gaussian curve was fit onto the main peak of ITD tuning curve, with the best ITD determined by the maximum of the Gaussian. The main peak was defined as the peak in the ICx curve with the largest response as the side peaks in ICx are usually much smaller relative to the main peak (*Figure 2*). In ICCls, as there is no side peak suppression due to the narrow frequency tuning (*Wagner et al., 1987*; *Fujita and Konishi, 1991*), the peak corresponding to the neurons' characteristic delay was determined during the recording by comparing neurons along the dorsal-ventral axis, which share

a characteristic delay but show different frequency tuning, which manifests in neurons along this axis sharing one peak response at their characteristic delay, with other peaks misaligned (*Wagner et al., 1987*; *Fujita and Konishi, 1991*; *Wagner et al., 2007*). Thus, the peak corresponding to preferred ITD was defined ad hoc for each recording and ITDs not comprising the main peak were ignored for fitting the Gaussian.

As previously reported (*Cazettes et al., 2014*; *Cazettes et al., 2016*), the frequency tuning range was defined as the range of frequencies that elicited more than 50% of the maximum response. The lower and upper edges of this range correspond to the lowest and highest frequencies of the response range. Because ICx frequency tuning curves are generally broad and without a single peak, the best frequency was defined as the mean of the lowest and highest frequencies of the frequency range.

Frontally tuned neurons were defined as neurons with a best ITD between ±30 µs, which equates to ±10°. While it is possible that neurons with this tuning could instead be tuned to rearward directions, this is extremely unlikely, and steps were taken to ensure this did not occur. Firstly, as the ICx is topographic, the best ITD increases with eccentricity. The most peripheral regions of the map are not known to respond to small ITDs, suggesting that rearward locations are not represented. This is consistent with the ICx projecting topographically onto the optic tectum and vice versa (*Hyde and Knudsen, 2000*; *DeBello et al., 2001*), which is multimodal and would not need to accurately represent rearward space in a binocular species. In addition, visual information from the optic tectum calibrates the representation of auditory space in the ICx (*Knudsen and Knudsen, 1989*; *Brainard and Knudsen, 1993*; *Gutfreund et al., 2002*), limiting this possibility. The spatial tuning of optic tectum neurons that do respond to rearward locations are extremely broadly tuned, consistent with this notion (*Knudsen, 1982*). Additionally, as multiple penetration sites were used during a given surgery, the location of the recording site in the topographic map could be tracked; no unreasonably large changes in ITD tuning across nearby penetration sites were detected.

Pearson correlation coefficients were calculated for each experimental group, pooled across birds. Student's *t*-test was used to compare regression slope values. Mann–Whitney *U*-test was used to compare frequency tunings for frontal neurons between groups. Effect sizes were derived from the rank-biserial correlation, which uses the *U* statistic to compute the overlap between groups by calculating the correlation between the data and its rank in the Mann–Whitney *U*-test, on a scale of 0–1 (1 indicating no overlap).

## Acknowledgements

This study was supported by NRSA F31 DC019303 to KS; DC007690 and NS104911 (Brain Initiative). We thank Hermann Wagner for providing HRTFs datasets used for this work, and Andrea Bae, Roland Ferger, and Brian Fischer for feedback and comments on data analysis and manuscript.

## Additional information

### Funding

| Funder | Grant reference number | Author |
| --- | --- | --- |
| National Institute on Deafness and Other Communication Disorders | F31DC019303 | Keanu Shadron |
| National Institute on Deafness and Other Communication Disorders | R01DC007690 | José Luis Peña |
| National Institute of Neurological Disorders and Stroke | R01NS104911 | José Luis Peña |

The funders had no role in study design, data collection and interpretation, or the decision to submit the work for publication.

## Author contributions
Keanu Shadron, Conceptualization, Data curation, Formal analysis, Funding acquisition, Validation, Investigation, Visualization, Methodology, Writing – original draft, Writing – review and editing; José Luis Peña, Conceptualization, Resources, Supervision, Funding acquisition, Writing – original draft, Project administration, Writing – review and editing

## Author ORCIDs
Keanu Shadron ID http://orcid.org/0000-0001-5217-9282

## Ethics
All procedures complied with the National Institute of Health guidelines and were approved by the institutional animal care and use committee of the Albert Einstein College of Medicine (00001189). All surgeries were performed under ketamine/xylazine to minimize suffering. The trimming of facial ruff feathers were painless and did not require anesthetics. No experiments were terminal.

## Decision letter and Author response
Decision letter https://doi.org/10.7554/eLife.84760.sa1
Author response https://doi.org/10.7554/eLife.84760.sa2

# Additional files

## Supplementary files
• MDAR checklist

## Data availability
All data and original code used in this paper is publicly available at on G-Node.

The following dataset was generated:

| Author(s) | Year | Dataset title | Dataset URL | Database and Identifier |
| --- | --- | --- | --- | --- |
| Shadron K, Peña JL | 2022 | Shadron_Pena_2022 | https://doi.org/10.12751/g-node.p52i4s | G-Node, 10.12751/g-node.p52i4s |

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
