## [Editor Report]

This research advance shows that if juvenile barn owls experience experimentally altered interaural time differences – the binaural cue used for localizing sounds in the horizontal plane – the frequency tuning properties of neurons in the space-mapped region of the midbrain undergo adaptive changes. The results therefore suggest that the statistics of sound stimulation can influence the sensitivity of auditory midbrain neurons to a fundamental stimulus feature in the developing barn owl brain. These findings will be of interest to the fields of developmental and sensory neuroscience.

---

## [Decision Letter]

**Decision letter after peer review:**

Thank you for submitting your article "Development shaped by cue reliability in the barn owl's auditory midbrain" for consideration by *eLife*. Your article has been reviewed by 2 peer reviewers, and the evaluation has been overseen by Andrew King as the Senior and Reviewing Editor. The following individual involved in the review of your submission has agreed to reveal their identity: Michael Pecka (Reviewer #1).

Essential revisions:

1. The paper does not establish a causal link between ITD reliability and the observed physiological changes. If the key conclusions of the paper are to be retained, stronger evidence needs to be provided that changes in the response properties of midbrain neurons can be accounted for by frequency-dependent changes in the reliability of this localization cue. If not, the authors should consider alternative explanations.

2. A more comprehensive analysis of the acoustical effects of facial ruff removal and the extent to which this leads to changes in ITD reliability is needed since the reviewers had reservations about the robustness and ecological validity of the ITD reliability metric used. See the comments of reviewer 2 for more details on this point.

3. ITD sensitivity at frequencies other than the neuronal best frequency needs to be considered, in particular, to demonstrate whether the neurons show ITD sensitivity at higher frequencies. This is particularly an issue given the changes in the tuning width of the neurons.

*Reviewer #1 (Recommendations for the authors):*

I find the title difficult to interpret: development of what?

Abstract: Can sensory systems "choose" stimuli? Do you mean "are more responsive" or are you referring to perceptual levels?

Line 80f: Is this sentence correct? Shouldn't it read "ICx neurons of juvenile and ruff-removed….were predominately tuned to frequencies lower than… In normal adults"?

Can the authors explain why a correlation between ITD and ILD tuning emerges as a function of ruff removal?

*Reviewer #2 (Recommendations for the authors):*

L22: statistics of spatial cues are likely to vary over time, so I suspect what is meant here is a 'long-term average' rather than permanent.

L45: it would be helpful to clearly identify where this definition of ITD reliability comes from and refer to possible alternatives. Also, the corruptibility of ITD cues is presumably not fully determined by the HRTFs but by the acoustical properties of the environment. Finally, the key variable of interest is presumably the spatial reliability of ITD cues, which may not be fully determined by ITD reliability.

L80: how are frontally-tuned neurons distinguished from neurons tuned to rearward locations?

L86: in conjunction with what is said elsewhere, this implies that high-frequency ITD sensitivity is not preserved in the ICx, but it is not clear what data support this claim.

L113: although IPD reliability may be a factor in determining ITD reliability, the two are not necessarily the same.

L124: it looks like there are changes in IPD reliability for low frequencies at ~160 degrees (i.e. behind the animal), but these do not seem to be obvious in Figure 1C.

L126: what precise azimuth values were used? Referring to 45 degrees makes it seem as if a 5-degree resolution was used, but the methods suggest 20 degrees.

Figure 1: it would be helpful to mark the azimuth values corresponding to the precise values tested. Also, it should be stated clearly if the data have been averaged for left and right hemifields (or whether only one hemifield is being shown).

L127: if I have understood correctly, the measures in Figure 1 are partly determined by the gain associated with rearward locations. If so, is there a uniform decrease in gain for all frequencies at all locations following facial ruff removal? If not, what impact might this have on ITD reliability?

L131: do the data also suggest the opposite for rearward locations (~160 degrees)? Also, does this predicted association between best frequency and ITD reliability assume a linear weighting of ITDs determined by the frequency tuning function? And if so, is this assumption true of ICX neurons?

P4, paragraph 3, line 4: are neurons with a best ITD of 0 necessarily tuned to frontal locations or could they be tuned to rearward (or midline) locations?

L189: this seems to imply that neurons can only represent ITDs at their best frequency. Could the ICx preserve sensitivity to ITDs at high frequencies using neurons that have slightly lower best frequencies?

L197: it is unclear why there is a focus on frontally-tuned neurons here (and only frontal locations). Figure 3 shows changes in neurons with best ITDs > 150 microseconds and Figure 1 shows changes in IPD reliability for rearward locations.

L205-207: 'ability to use high frequencies' is confusing since there is no behavioural measure. I'm not entirely sure what this means above and beyond what was already stated about the azimuthal spatial tuning.

L210: If I've understood correctly, these results suggest that facial ruff removal does not affect ITD cues for azimuthal locations within 60 degrees of the midline and, if anything, enhance ILD cues available for these locations (although this would be clearer if the acoustical cues themselves were plotted). However, they do not say what happens to ITDs and ILDs (or overall gain) for rearward locations, which appear to contribute to the IPD variability measures used (even for frontal locations). Also, if facial ruff removal makes ILD cues available for azimuthal localization, then this may affect the need to use ITDs at high frequencies, even if IPD reliability is unchanged. Some discussion of these issues would therefore be helpful.

P7, line 1: 'frontal neurons' seems to refer to neurons with a best ITD of 0, but could they be tuned to rearward locations (or to midline locations)?

L322: I didn't fully understand how these results show that ITD reliability has already begun to refine neural tuning. I would have thought longitudinal recordings would be necessary to establish this, but perhaps I misunderstood what was intended.

L346: following facial ruff removal, is the change in gain equal for all frequencies at all locations (including those to the rear)? If facial ruff removal increases the gain for high-frequency sounds behind the animal, what would this do to measures of IPD reliability?

L350: is the rate of change for ITD unaffected by ruff removal for all locations (including rearward locations)?

L354: how would these analyses differ if rearward locations were included and changes in gain were taken into account?

L393: is this true of all locations, or is it limited to frontal locations?

L395: are the results of that study consistent with the present study, or are there differences between the midbrain and forebrain?

L414: I found it confusing to refer to 'location' here when what I think is meant is the IPD associated with the peak of the cross-correlation function.

L415: IPD and ITD are sometimes used interchangeably in the manuscript, but it would be helpful to distinguish between them clearly since a given change in IPD does not always correspond to the same change in ITD.

L432: 'full spatial tuning map' seems to imply that this analysis was done for all locations (including those to the rear), but this is not plotted in the relevant figure.

L465: additional details (or references) should be provided to indicate whether all stimuli (including tones) were binaural and how ILDs were added to the stimulus (e.g. was level fixed in one ear etc.).

L474: is it possible that facial ruff removal might change this?

---

## [Author Response]

Essential Revisions:

1. The paper does not establish a causal link between ITD reliability and the observed physiological changes. If the key conclusions of the paper are to be retained, stronger evidence needs to be provided that changes in the response properties of midbrain neurons can be accounted for by frequency-dependent changes in the reliability of this localization cue. If not, the authors should consider alternative explanations.

Significant additions to the manuscript, described throughout this letter, provide further controls to support the causal link between ITD reliability and frequency tuning in the ICx. Briefly, new HRTF analysis suggests that the described pattern of ITD reliability is stable across acoustical environments (Figure 1—figure supplement 1). We also performed additional gain analysis (new Figure 5) to provide evidence that ITD reliability is the principal factor driving frequency tuning, as opposed to other potential factors. Finally, further analyses were conducted and reported in this letter (denoted with an ‘L’ in figure numbers) to indicate the full extent of the shift in frequency tuning observed following ruff-removal. While we felt that these analyses could address Reviewers’ specific concerns but might not be necessary for this report, they can be added to the manuscript if deemed useful. Overall, we consider these additions provide supplementary evidence that the changes in frequency tuning can be attributed to the change in ITD reliability.

2. A more comprehensive analysis of the acoustical effects of facial ruff removal and the extent to which this leads to changes in ITD reliability is needed since the reviewers had reservations about the robustness and ecological validity of the ITD reliability metric used. See the comments of reviewer 2 for more details on this point.

Additional analyses on the effect that ruff-removal had on gain were added to the manuscript (Figure 5), indicating that the change in gain does not explain the change in frequency tuning observed in the ICx. To address the Reviewer’s concerns of ecological validity, we also performed additional analyses where we measured ITD reliability under different conditions: in the case of maskers at 50% the power of the targets, and where target and masker sounds were natural recordings of prey noises (a mouse scream). Using these parameters resulted in qualitatively similar patterns of ITD reliability, suggesting that this metric is robust across sound conditions. One note is that while this analysis requires sounds to be relatively broadband, but because the sound is decomposed into individual frequency channels by the cochlea (Kӧppl, 1997), this analysis would be consistent across naturalistic sounds.

3. ITD sensitivity at frequencies other than the neuronal best frequency needs to be considered, in particular, to demonstrate whether the neurons show ITD sensitivity at higher frequencies. This is particularly an issue given the changes in the tuning width of the neurons.

To address the Reviewers’ concerns, we conducted new analysis and electrophysiology recordings. We first assessed the normalized frequency tuning curves of all the frontally-tuned neurons of the ruff-removed owls (Author response image 1). The average frequency tuning across these neurons indicates that these neurons respond very little to frequencies outside their frequency range, near their basal response level. This would make it very unlikely that frontal ICx neurons in ruff-removed owls could represent ITD accurately for high frequencies outside their responsive frequency range.

**Author response image 1. sa2fig1:** Frequency tuning of frontally-tuned ICx neurons in ruff-removed owls. Tuning curves are normalized by the max response. Thick black line indicates the average tuning curve. Dashed black line indicates basal response.

In addition, we also performed new recordings to measure ITD sensitivity across frequencies in a ruff-removed owl (the other ruff-removed owl laid eggs just after the Reviews). Two example neurons recorded in this owl are shown in Author response image 2. We recorded ITD tuning curves for tones (colored lines) across the owl’s hearing range, as well as for broadband noise (solid black line). These responses indicate that for tones within the neuron’s frequency range (colored solid lines), neurons showed ITD sensitivity, with clear peaks for their best ITD and side-peak responses. However, for tones outside of the frequency range (colored dotted lines), responses were highly attenuated, and there was no clear ITD tuning. For example, in both neurons, the tuning curve for 7 kHz (yellow) in both neurons was largely flat across ITDs. Combined, these analyses suggest that there is little responsiveness, and thus little ITD sensitivity, for frequencies outside of a neuron’s frequency range. Because the ICCls, as well as the other upstream regions in the ITD-detection circuit, are narrowly tuned to frequency, this supports our premise that if an ICx neuron is not tuned to a given frequency, then the neuron is not receiving strong enough ITD information for this frequency.

**Author response image 2. sa2fig2:** ITD sensitivity across frequencies in ruff-removed owl. Two example neurons shown in a and b. ITD tuning for tones (colored) and broadband (black) plotted by firing rate (non-normalized). Solid colored lines indicate responses to frequencies that are within the neuron’s preferred frequency range (i.e. above the half-height, see Methods), dashed lines indicate frequencies outside of the neuron’s frequency range.

We were unsure whether the mentioned ‘tuning width’ was referring to frequency or ITD tuning. If the Reviewers meant frequency tuning width: we did not find any difference in frequency tuning width between the normal and ruff-removed owls (manuscript Figure 8c). Because of this, neurons in ruff-removed owls should not be any more responsive to non-preferred frequencies than neurons in normal owls. If Reviewers instead meant ITD tuning width: ITD tuning width is negatively correlated with and determined by stimulus frequency (and thus the frequencies a given neuron responds to) in the IC of barn owls (Takahashi and Konishi, 1986; Wagner et al., 2002; Cazettes et al., 2016) and mammals (McAlpine et al., 2001; Yin and Kuwada, 1983). Thus, the ITD tuning width at a given frequency (e.g. 6 kHz) would not differ for normal or ruff-removed ICx neurons, but differs for broadband noise depending on frequency tuning.

Reviewer #1:I find the title difficult to interpret: development of what?

Regarding this reviewer’s recommendation, the title was amended to specify that effect of ITD reliability on development of frequency tuning was investigated in this study and all suggested text changes regarding sentence structure were recognized and corrected, including a clearer explanation on why ILD becomes correlated with ITD following ruff-removal added to the text accompanying Figure 2 (Lines 139-143).

Reviewer #2:L22: statistics of spatial cues are likely to vary over time, so I suspect what is meant here is a 'long-term average' rather than permanent.

That is correct and has been reworded.

L45: it would be helpful to clearly identify where this definition of ITD reliability comes from and refer to possible alternatives. Also, the corruptibility of ITD cues is presumably not fully determined by the HRTFs but by the acoustical properties of the environment. Finally, the key variable of interest is presumably the spatial reliability of ITD cues, which may not be fully determined by ITD reliability.

A more thorough description of ITD reliability was added, including reference to previous studies that have used the same definition (Lines 55-60, 100 and 371-374). While the acoustical properties of the environment would clearly have an effect on reliability, they are considered outside the scope of this work, which has used a manipulating test assumed to affect acoustic properties of the head (HRTF), rather than the environment. However, we note that the facial ruff, ear canals and head finish growing before two months of age, while they remain dependent on the parents for at least one more month (Haresign and Moiseff, 1988). This may suggest that the owl’s early auditory experience is largely uniform in the nest, and environmental effects on ITD reliability may play a smaller role in this developmental stage.

L80: how are frontally-tuned neurons distinguished from neurons tuned to rearward locations?

This was described in the Methods section (Lines 468-479).

L86: in conjunction with what is said elsewhere, this implies that high-frequency ITD sensitivity is not preserved in the ICx, but it is not clear what data support this claim.

This was addressed above in our response to essential revisions indicated by editors.

L113: although IPD reliability may be a factor in determining ITD reliability, the two are not necessarily the same.

This was addressed above in our response to essential revisions indicated by editors.

L124: it looks like there are changes in IPD reliability for low frequencies at ~160 degrees (i.e. behind the animal), but these do not seem to be obvious in Figure 1C.

This is likely due to a very small decrease in IPD reliability around 3 kHz at 160°, which affects across-frequency normalization. However, there have been no neurons reported in the literature tuned to such rearward space, so are considered unlikely to be specifically tested in this study.

L126: what precise azimuth values were used? Referring to 45 degrees makes it seem as if a 5-degree resolution was used, but the methods suggest 20 degrees.Figure 1: it would be helpful to mark the azimuth values corresponding to the precise values tested. Also, it should be stated clearly if the data have been averaged for left and right hemifields (or whether only one hemifield is being shown).

As described in the Methods, 20° steps were used. We realize the discrepancy with Figure 1 and have changed the tick values to more accurately represent the step size, which we note it did not affect results. Finally, we denote in the Figure 1 legend that the right hemifield is shown. Both hemifields are highly similar, so only one was chosen for clarity. Alternatively, an average of the two hemifields could be performed as well, if considered necessary.

L127: if I have understood correctly, the measures in Figure 1 are partly determined by the gain associated with rearward locations. If so, is there a uniform decrease in gain for all frequencies at all locations following facial ruff removal? If not, what impact might this have on ITD reliability?

A new figure (new Figure 5) was added detailing changes in gain following ruff-removal. The results of this analysis, consistent with von Campenhausen et al., 2006, show a uniform decrease in the gain across frequencies. Thus, the largest gain is seen for high frequencies from frontal locations, in both the normal and ruff-removed conditions. This indicates that the change in gain alone does not fully reflect the observed changes in frequency tuning, nor the pattern of frequency tuning across locations (also shown in Cazettes et al., 2014).

L131: do the data also suggest the opposite for rearward locations (~160 degrees)? Also, does this predicted association between best frequency and ITD reliability assume a linear weighting of ITDs determined by the frequency tuning function? And if so, is this assumption true of ICX neurons?

While ruff-removal induces a strong change in ITD reliability of high frequencies at frontal locations, differences in the pattern of ITD reliability at rear locations did not increase for high frequencies. As such, we would not expect to see any change in frequency tuning for neurons tuned to rearward locations. Text addressing this was added to the discussion (Lines 313-321)

P4, paragraph 3, line 4: are neurons with a best ITD of 0 necessarily tuned to frontal locations or could they be tuned to rearward (or midline) locations?

This was addressed in our response to the essential revisions indicated by editors.

L189: this seems to imply that neurons can only represent ITDs at their best frequency. Could the ICx preserve sensitivity to ITDs at high frequencies using neurons that have slightly lower best frequencies?

This was also addressed in our response to the essential revisions indicated by editors.

L197: it is unclear why there is a focus on frontally-tuned neurons here (and only frontal locations). Figure 3 shows changes in neurons with best ITDs > 150 microseconds and Figure 1 shows changes in IPD reliability for rearward locations.

This is largely addressed in the essential revisions done by editors’ recommendations. In addition, the peripheral neurons recorded in the ruff-removed owls, with best ITDs between ~150-250 µs, correspond to ~40-75° in eccentricity (von Campenhausen et al., 2006). At these locations, these neurons’ frequency tunings still align with the most reliable frequencies (Figure 1b). Additional text was added (Lines 155-157) to clarify the relationship between ITD and azimuth, where ITD varies linearly with azimuth at approximately 3 µs/deg in the ruff-removed condition (Knudsen et al., 1994; von Campenhausen et al., 2006).

L205-207: 'ability to use high frequencies' is confusing since there is no behavioral measure. I'm not entirely sure what this means above and beyond what was already stated about the azimuthal spatial tuning.

We acknowledged this and adjusted the wording (Lines 189-190).

L210: If I've understood correctly, these results suggest that facial ruff removal does not affect ITD cues for azimuthal locations within 60 degrees of the midline and, if anything, enhance ILD cues available for these locations (although this would be clearer if the acoustical cues themselves were plotted). However, they do not say what happens to ITDs and ILDs (or overall gain) for rearward locations, which appear to contribute to the IPD variability measures used (even for frontal locations). Also, if facial ruff removal makes ILD cues available for azimuthal localization, then this may affect the need to use ITDs at high frequencies, even if IPD reliability is unchanged. Some discussion of these issues would therefore be helpful.

As the overall effects of the ruff-removal on ITDs and ILDs were extensively characterized in von Campenhausen et al., 2006, which these HRTFs are from, we did not deem it worthwhile to address all of these. Nevertheless, these effects are mentioned throughout the manuscript, and additional analysis on gain was added to the manuscript (Lines 168-179).

P7, line 1: 'frontal neurons' seems to refer to neurons with a best ITD of 0, but could they be tuned to rearward locations (or to midline locations)?

This was addressed in the essential revisions suggested by editors.

L322: I didn't fully understand how these results show that ITD reliability has already begun to refine neural tuning. I would have thought longitudinal recordings would be necessary to establish this, but perhaps I misunderstood what was intended.

We realized this confusion and altered the text accordingly (Line 262). We do not believe ITD reliability is already refining neural tuning at this time point, given our current data.

L346: following facial ruff removal, is the change in gain equal for all frequencies at all locations (including those to the rear)? If facial ruff removal increases the gain for high-frequency sounds behind the animal, what would this do to measures of IPD reliability?

This was addressed in the new analysis and Figure 5. The loss in gain is consistent across frequencies, with an increase in gain for the most rearward locations at high frequencies. As described in the Methods, gain is part of the analysis of IPD reliability, so the results in Figure 1 already take this change in gain into account.

L350: is the rate of change for ITD unaffected by ruff removal for all locations (including rearward locations)?

The rate of change for ITD for peripheral space is sharper than frontal space in the normal condition. After ruff-removal, the rate of change is unaffected for frontal space, but decreases for peripheral and rearward space, becoming equal to frontal space at 3 µs/deg (Von Campenhausen et al., 2006).

L354: how would these analyses differ if rearward locations were included and changes in gain were taken into account?

The analysis in Figure 7 already took rearward locations into account, but the axes were limited to ±60° for brevity. We updated this figure to include ±160°, which did not affect the conclusions of the analysis. Additionally, gain is implicitly included in the analysis of ILD spatial tuning.

L393: is this true of all locations, or is it limited to frontal locations?

While the largest change in ITD reliability in our data is confined to frontal locations, we do not expect large differences in the neural architecture between frontally-tuned and peripherally-tuned regions of the ICCls/ICx. Previous reports have shown uniform plasticity across the map, supporting this premise (reported by Nichols and DeBello, 2008).

L395: are the results of that study consistent with the present study, or are there differences between the midbrain and forebrain?

Text was revised to indicate that the results are consistent (Line 353).

L414: I found it confusing to refer to 'location' here when what I think is meant is the IPD associated with the peak of the cross-correlation function.

We understand this confusion and revised ‘location’ to ‘point’ (Line 375).

L415: IPD and ITD are sometimes used interchangeably in the manuscript, but it would be helpful to distinguish between them clearly since a given change in IPD does not always correspond to the same change in ITD.

This was addressed above, in our response to Reviewer #2’s Public review comments.

L432: 'full spatial tuning map' seems to imply that this analysis was done for all locations (including those to the rear), but this is not plotted in the relevant figure.

As described above in the response to comments about L354, we already performed this analysis, but had not included it in the original version of Figure 7. We have updated this figure to include rearward space.

L465: additional details (or references) should be provided to indicate whether all stimuli (including tones) were binaural and how ILDs were added to the stimulus (e.g. was level fixed in one ear etc.).

Text was added to the Methods (Lines 431-433) to indicate that all stimuli were binaural and how ILD was computed.

L474: is it possible that facial ruff removal might change this?

Extensive studies regarding the plasticity of the barn owl’s auditory space map note shifts or adjustments to the space map, but topography remains clear, given the feedback from the retinotopic map (Brainard and Knudsen, 1993; Brainard and Knudsen, 1998; Hyde and Knudsen, 2000; Gutfreund et al., 2002). Previous work using ruff-removed owls did not note changes in the topographic tuning in the auditory midbrain (Knudsen et al., 1994). Therefore, we do not expect increased heterogeneity in the tuning of nearby ICx neurons after ruff-removal.